# Accelerated Fractionated Radiation Therapy for Localized Glottic Carcinoma

**Tatsuji Mizukami [1,\*]** , **Kentaro Yamagishi [1]** , **Masaki Tobikawa [1]** , **Akira Nakazato [2]** , **Hideharu Abe [2]** , **Yuka Morita [2]** and **Jun-ichi Saitoh [1]**

1  Division of Radiation Oncology, Department of Radiology, Faculty of Medicine, Academic Assembly, University of Toyama, Toyama 930-0194, Japan; junsaito@med.u-toyama.ac.jp (J.-i.S.)
2  Department of Otorhinolaryngology, Head & Neck Surgery, University of Toyama, Toyama 930-0194, Japan
\*  Correspondence: tmizukam@med.u-toyama.ac.jp; Tel.: +81-76-434-7326

**Abstract:** Background: The aim of this study is to examine the outcomes of an accelerated fractionated irradiation for N0 glottic carcinoma. Methods: In this retrospective analysis, 29 patients with N0 glottic carcinoma treated by radiation therapy were enrolled. Thirteen patients had T1a disease, six had T1b disease, and ten had T2 disease. A fractional dose of 2.1 Gy was administered to seven patients. The total doses were 65.1 and 67.2 Gy in four and three patients, respectively. A fractional dose of 2.25 Gy was administered to 22 patients. The total doses were 63 and 67.5 Gy in 21 patients and 1 patient with T2 disease, respectively. Additionally, 13 patients underwent the use of TS-1 (80–100 mg per day). Results: The median follow-up period was 33 months, and the 3-year local control rate was 95.6%. No patient had a lymph node or distant recurrence. As acute adverse events, grades 2 and 3 dermatitis were observed in 18 patients and 1 patient, and grades 2 and 3 mucositis were observed in 15 patients and 1 patient. As a late adverse event, one patient required tracheotomy because of laryngeal edema occurring. Conclusions: Accelerated fractionated irradiation may be an option in the radiation therapy of N0 glottic carcinoma because of its ability to shorten the treatment time.

**Keywords:** glottic carcinoma; accelerated fractionated radiation therapy; laryngeal cancer; fractional dose





## 1. Introduction

Laryngeal cancer is a significant clinical problem. In 2019, 5111 people were diagnosed with the disease in Japan, and in 2020, the number of deaths from the disease was 781. The number of deaths from the disease has remained flat, but the number of people affected by the disease is increasing every year [1]. It is estimated that 13,430 people in the United States were diagnosed with laryngeal cancer in 2016, and 3620 people died from the disease. Laryngeal cancer accounts for only 0.8% of all newly diagnosed cancers in the United States annually. However, the larynx is an organ that plays an important role in voice production and swallowing, and when it is affected, the patient's quality of life is greatly impaired. Thus, it has important social significance [2]. Glottic carcinomas represent 66% of all laryngeal cancers, is more common in men, and is closely related to smoking. A Brinkman index (number of cigarettes smoked per day × number of years smoked) of 600 or more is considered a high-risk group. The peak age groups affected by this cancer are those in their 60s and 70s, followed by those in their 50s and 80s. In most patients with glottic carcinoma, the histopathological diagnosis is frequently squamous cell carcinoma, and lymph node metastasis rarely occurs. In addition, glottal carcinoma is more common with highly differentiated carcinoma and less common with poorly differentiated carcinoma. Glottic carcinoma is easily detected at an early stage because hoarseness appears as a symptom from an early stage. To preserve the various functions of the larynx, including voice and speech functions, radiotherapy is the first choice for early-stage cancer [3,4].

The usual irradiation dose is 2 Gy per fraction using 4–6 MV X-rays in a standardized field from the right and left directions, with a total dose of approximately 60–66 and 70 Gy for T1 and T2 diseases, respectively, and the 5-year local control rate of radiation therapy alone is 77–94% and 70–80% at T1 and T2, respectively [5].

The modification of irradiation schedules, the modification of irradiation methods, or combination chemotherapy are being attempted to improve treatment outcomes. An accelerated fractionated radiation therapy (AFRT), in which a fractional dose is increased in anticipation of a shorter completion time and better local control, has also been attempted [6], and a dose of 2.25 Gy per fraction is widely used worldwide. In Japan, Yamazaki et al. reported that better local control was obtained in the group receiving radiation therapy at a dose of 2.25 Gy for stage 1 glottic carcinoma [7]. Contrarily, the JCOG0701 clinical trial did not prove the non-inferiority of the 2.4-Gy dose in terms of the progression-free survival rate, although the local control rate and adverse events were comparable [8].

At our hospital, accelerated fractionation has been used for glottic carcinomas since 2006, and we report here the efficacy and adverse events of AFRT.

## 2. Materials and Methods

The patients in the present retrospective study had localized stage glottic carcinomas and underwent radiation therapy at our hospital between January 2010 and March 2019. There were 29 patients, including 13 patients with T1a-stage disease, 6 with T1b-stage disease, and 10 with T2-stage disease. No lymph node or distant metastasis was noted in any of the patients. Only one patient was female. The patients' ages ranged from 43 to 92 years, with a median age of 70 years. Laryngoscopy and biopsy were performed, and in all patients, the histological diagnosis was squamous cell carcinoma. All patients were untreated and had no history of radiation therapy to the neck (Table 1).

**Table 1.** Patient and tumor characteristics.

| Characteristic | |
|---|---|
| **Sex** | |
| Male/female | 28/1 |
| T classification | |
| T1a | 13 |
| T1b | 6 |
| T2 | 10 |
| Age (years) | |
| Median | 70 |
| Min–Max | 43–92 |

Written consent for treatment was obtained from all patients. The study protocol was approved by the Institutional Review Board of Toyama University Hospital.

Adverse events were evaluated using the Common Terminology Criteria for Adverse Events version 5.0 [9].

A thermally variable shell was used in all patients, and a three-dimensional conformal radiation therapy (3D-CRT) plan was developed using a CT simulator. The treatment device used was a Clinac 21EX linear accelerator (Varian Medical Systems, Palo Alto, CA, USA).

The patients were treated with 6-MV X-rays at the right and left contralateral portals. The vocal cords were defined as the clinical target volume (CTV), and the center of the CTV was set as the center of the irradiation field. The irradiation field's upper border was set at the inferior border of the hyoid bone, and the posterior border was the anterior border of the vertebral body. The anterior area was defined as the area that adequately included the body surface. The irradiated field size varied from 4.4 cm × 3.7 cm to 6 cm × 6 cm, with most patients (10) having an irradiated field size of 5 cm × 5 cm. Wedges of none, 15 degrees, and 30 degrees were used in 6, 17, and 6 patients, respectively.

A fractional dose of 2.1 Gy was administered to seven patients. The total doses were 65.1 and 67.2 Gy in four and three patients, respectively. A fractional dose of 2.25 Gy was administered to 22 patients. The total dose was 63 Gy in 21 patients and 67.5 Gy in one patient with T2 disease. Additionally, 13 patients treated before 2016 underwent the concurrent use of TS-1 (formulated by Taiho Pharmaceutical, Tokyo, Japan, approved by the Japanese government). The dose of TS-1 was 80 mg per day in five patients and 100 mg per day in the other eight patients. The patients received TS-1 for two weeks at the same time as the start of radiotherapy and for two more weeks after a one-week pause. And 12 of these patients were also simultaneously treated with 30,000 units of vitamin A daily for five consecutive weeks (Table 2).

**Table 2.** Treatment and clinical outcomes.

| Treatment and Clinical Outcomes | | | | | |
|---|---|---|---|---|---|
| | n | Gy/fraction | Total Dose | TS-1 | Local recurrence |
| T1 | 2 | 2.1 | 65.1–67.2 | 2 | 0 |
| | 17 | 2.25 | 63 | 3 | 0 |
| T2 | 5 | 2.1 | 65.1–67.2 | 5 | 0 |
| | 5 | 2.25 | 63–67.5 | 3 | 1 |

The duration of treatment ranged from 38 to 50 days, with a median of 41 days.

## 3. Results

### 3.1. Response and Follow-Up

The median follow-up period was 33 months. Twenty-seven patients survived, none died of primary disease, and one patient died of other causes, including gastric cancer and renal failure. One patient had local recurrence, and the 3-year local control rate was 95.6%. The Kaplan–Meier survival curve is shown in Figure 1. At 36 months, both the local control and recurrence-free survival rates were 95.6%, and the overall survival rate was 100%.

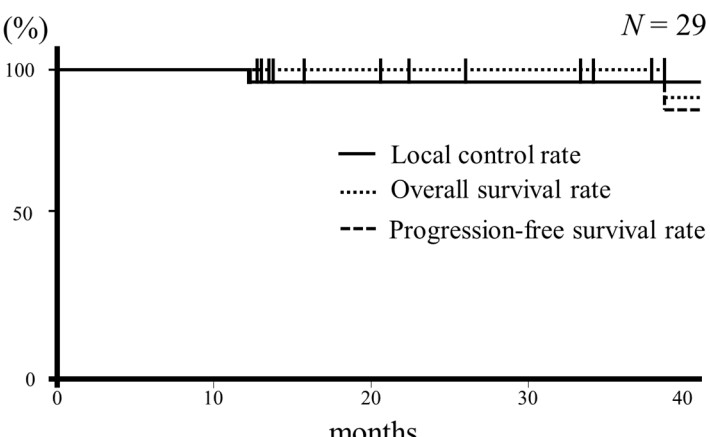

**Figure 1.** Kaplan–Meier estimates of the local control (solid line), overall (dotted line), and progression-free survival (dashed line) rates from the start of treatment.

Local recurrence was observed in a patient with T2 disease who received a total dose of 63 Gy with a fractional dose of 2.25 Gy. TS-1 was also used in combination. At 12 months after the treatment initiation, an ulcer was noted near the anterior commissure, and a biopsy revealed the presence of a recurrence. The depth from the body surface to the dosimetry point was slightly shallow (2.9 and 2.6 cm), and wedges were not used. This patient was subsequently treated with CyberKnife (Accuray, Sunnyvale, CA, USA), additional chemotherapy, and immune checkpoint inhibitors.

*3.2. Toxicity*

As acute adverse events, grades 2 and 3 dermatitis were observed in 18 patients and 1 patient, respectively. Grade 3 dermatitis was observed in a patient with T2 disease who was administered a total dose of 65 Gy along with TS-1. Grades 2 and 3 mucositis were observed in 15 patients and 1 patient, respectively.

As a late adverse event, one patient had worsening hoarseness after 6 months of treatment, and tracheotomy was performed after careful judgment by the attending physician. This patient received fractional and total doses of 2.25 and 63 Gy, respectively, with TS-1 administration. Acute grade 2 mucositis was observed in this patient, and the maximum dose on the dose distribution was approximately 104% (Table 3).

**Table 3.** Toxicity.

| Toxicity | | | | | |
|---|---|---|---|---|---|
| | Grade | 1 | 2 | 3 | 4 |
| Acute | Dermatitis | 10 | 18 | 1 | 0 |
| | Mucositis | 13 | 15 | 1 | 0 |
| Late | Laryngeal edema | 0 | 0 | 0 | 1 |

## 4. Discussion

In the present study, the 3-year local control rate for stage T1–2 glottic carcinomas treated with AFRT was 95.6%. Various centers have previously reported a fractional dose of 2 Gy and AFRT for glottic carcinomas. The 5-year local control rates after conventional fractionated radiation therapy (CFRT) alone were 77–94% and 70–80% for T1 and T2 diseases, respectively [5,10–15]. In Yamazaki et al.'s [7] randomized controlled trial of T1N0M0 glottic carcinoma patients treated with CFRT with a total dose of 60–66 Gy and AFRT with a fractional dose of 2.25 Gy and a total dose of 56.25–63 Gy, the 5-year local control rats were 77% and 92% in the CFRT and AFRT groups, respectively, and no significant difference in adverse events was observed between the two groups. Motegi et al. [16] reported the outcomes for glottic carcinoma treated with a single fractional dose of 2.4 Gy. According to the report, the 5-year local control and overall survival rates for the patients with T1 glottic carcinoma were 93% and 96%, respectively, and those for the T2 disease were 77% and 91%, respectively. No serious acute adverse events were observed, but serious late adverse events occurred in 1% of the patients. Compared to these previous reports, the results in this study seem to be comparable.

The results of JCOG0701, a randomized comparative study of CFRT versus AFRT with a fractional dose of 2.4 Gy in Japan, showed a 3-year progression-free survival rate of 79.9% for the CFRT group and 81.7% for the AFRT group, but the statistical superiority level was exceeded and non-inferiority was not proven. In a subanalysis, the 3-year survival rate in the 2-Gy group was higher than that previously reported, which may have had a statistical effect and prevented the non-inferiority of AFRT from being proven. Moreover, when the 3-year local control rate was calculated, the AFRT was considered better [8]. According to a previous prospective randomized trial of AFRT (KROG-0201; 63 Gy in 28 fractions for T1 and 67.5 Gy in 30 fractions for T2) versus CFRT in patients with T1–2 glottic carcinomas [6], the 5-year progression-free survival rate tended to be better in the AFRTgroup, but the difference was not statistically significant. The RTOG 9512 study conducted in the U.S., which compared hyperfractionated irradiation to CFRT, showed a 5-year local control rate of 78% in the hyperfractionated group receiving treatment at a dose of 1.2 Gy twice daily, but again, this was not a statistically significant improvement as compared to that of CFRT [17]. All randomized trials showed a trend toward better results with AFRT as compared to those of CFRT, but they did not show a statistically superior improvement.

In the past, a meta-analysis of radiotherapy in squamous cell carcinoma of the head and neck (MARCH) has been conducted. This analysis of radiotherapy fractionation

schedules included 33 clinical trials and 11,423 patients. These eligible trials and patients were divided into three pre-defined subgroups: hyperfractionation (eight comparisons: this is the group that received radiotherapy twice a day), moderately accelerated radiotherapy (21 comparisons: this is the group that was treated for about one week less than usual, and it also includes those with a single fraction dose of 2–2.5 Gy), and very accelerated radiotherapy (seven comparisons: this is the group for which the duration of treatment was less than half the usual duration). Clinical stages I–II were observed in 2922 cases, of which 2045 (70%) were laryngeal carcinoma. As a result, altered fractionated radiation therapy was associated with a significantly longer overall survival compared to conventional radiation therapy. However, there were differences among the three groups, the survival benefit being restricted to the hyperfractionated regimen group only. In terms of progression-free survival, all three groups showed superior improvement over radiation therapy in the conventional fractionation. The hyperfractionated regimen prevented both local and regional recurrence, while moderately accelerated radiotherapy was associated only with the prevention of local recurrence, but neither effect was observed with very accelerated radiotherapy. However, altered fractionation radiotherapy also increased adverse events. Compared to patients who received conventional radiation therapy, those who received altered fractionation radiotherapy had significantly increased rates of acute mucositis and the need for tube feeding during treatment. These results suggest that hyperfractionation therapy is best, but this method has not become the standard of care. The authors suggest that the main reasons for this are logistical issues, such as the difficulty of finding two slots per day on machines or patient management between fractions [18]. The treatment of our study is considered to fall under the category of moderately accelerated radiotherapy. The finding that moderately accelerated irradiation reduced local recurrence in the meta-analysis that included other head and neck cancers that are more difficult to treat than glottic carcinoma may support the benefit of AFRT in early-stage glottic carcinoma.

A more aggressive treatment other than changing the irradiation method may be the combination of chemotherapy. Saitoh et al. reported the results of T2 laryngeal cancer treated with CFRT at a dose of 66 Gy in 33 fractions plus daily low-dose cisplatin and weekly docetaxel. The 3- and 5-year overall survival rates were 95% and 93%, respectively, and the actuarial 3- and 5-year local control rates were both 94%. Grade 3 or higher hematologic toxicity and Grade 3 or higher mucositis were observed in 8% and 2% of patients, respectively [19]. On the other hand, Niibe et al. reported the results of concurrent RT at a dose of 60 Gy in 30 fractions or 61.2 Gy in 34 fractions combined with chemotherapy for T2N0 glottic carcinoma. A total of 83 patients were concurrently given UFT, 24 were given TS-1, 23 were given intravenous chemotherapy (mainly cisplatin), and 23 had no chemotherapy. According to the paper, the 3-year local control rates of the UFT, TS-1, and intravenous chemotherapy groups were 90.1%, 100.0%, and 73.4%, respectively. As adverse events, two patients in the UFT group experienced severe enteritis, but no patients in the other groups experienced severe adverse events [20]. Based on these results, Phase I and II trials of radiotherapy with TS-1 for patients with T2N0 glottic carcinoma were conducted at the authors' institution. The 3-year local control and overall survival rates were 94.7 and 85.4%, respectively. Mucositis was the most common adverse event and was observed in five (23%) patients at Grade 3, respectively [21]. Although the study was a retrospective analysis with a small number of cases, the results were as good as those of the present study. However, the occurrence of serious mucositis appears to be somewhat more common than in the present study. Increased toxicity with TS-1 concomitant therapy was also noted during the phase I study conducted by Tsuji et al. Four of the eighteen patients enrolled in the study developed grade 3 radiation dermatitis, which became dose-limiting toxicity at level 3 [22]. There have been other reports of good local control and survival after radiation therapy with TS-1 for glottic carcinoma, but all of them have small sample sizes [23–25]. In the present study, concurrent chemotherapy was used in approximately half of the patients. However, only one case of local recurrence was observed in a patient treated with AFRT with a total dose of 63 Gy and concurrent chemotherapy; thus, it is

unclear whether chemotherapy was beneficial. While AFRT alone has been reported to provide better local control than conventional radiotherapy alone [7], the RTOG0129 study reported that AFRT did not improve the prognosis compared to conventional radiotherapy when cisplatin was also used [26]. In the present study, accelerated irradiation caused late severe laryngeal edema. There have also been reports of severe mucositis and dermatitis with combined TS-1 irradiation, and there is great concern that the combination of the two may cause severe toxicity. Therefore, chemotherapy is not currently used in combination with AFRT at our hospital.

Today, the use of more precise radiation therapy techniques is being considered for various types of cancer, and this is also true for glottic carcinoma. Mohamed et al. reported oncologic outcomes after conventional radiotherapy using opposed lateral beams and intensity-modulated radiation therapy (IMRT) for T1N0 glottic carcinoma. The 3-year locoregional control for patients treated with conventional radiotherapy was 94%, compared to 97% with IMRT. The 3-year overall survival for patients with conventional radiotherapy was 92.5%, compared to 100% with IMRT. Post-radiation cerebrovascular events were in four patients in the conventional radiotherapy cohort, whereas no patients in the IMRT cohort suffered any events. Based on these results, the authors have transitioned the standard irradiation method to IMRT [27]. In addition, Kato et al. compared the dose distribution of IMRT and passive scattering proton therapy (PSPT) for early-stage glottic carcinoma and reported that PSPT can further reduce the dose to the organ at risk such as the carotid artery and spinal cord while maintaining the uniformity of the planning target volume dose distribution [28]. There is also the opinion that the standard-of-care radiation treatment of early-stage glottic carcinoma continues to be three-dimensional opposed lateral fields to include the whole larynx [29]. But clinical use has already begun and should be promptly validated in the future.

Although there have not been many studies examining the relationship between radiation therapy and genetic mutations, there have been recent attempts to use next-generation sequencers to generate gene profiles associated with the radiation therapy outcomes [30]. Although it may be difficult to perform such a technique because the biopsy specimens are often small, it may be possible to use a similar method to predict the efficacy of treatments for glottic carcinoma and to consider promoting personalized treatments, such as stronger treatments for patients who are thought to be resistant to treatment.

Our study has some limitations. It was a retrospective study with a small sample. Since the results were obtained only from a single institution, there may be bias in patient backgrounds and irradiation methods. However, both the present study and JCOG0701 trial showed an improvement over previous results, and it is possible that improved treatment accuracy, such as CT-based treatment planning and the use of fixtures, may have contributed to this improvement. If a treatment-resistant group is identified by the pathological background and genetic mutation information, the treatment intensity can be possibly increased for this group to further improve the treatment outcome.

## 5. Conclusions

In the present study, the 3-year local control rate was 95.6%, which was comparable to those of previous reports. AFRT may be an optional radiotherapy method for N0 glottic carcinomas because of its ability to shorten the treatment time.

**Author Contributions:** Conceptualization, T.M. and J.-i.S.; methodology, T.M. and J.-i.S.; formal analysis, T.M.; investigation, T.M.; data curation, T.M., K.Y., M.T. and J.-i.S.; writing—original draft preparation, T.M.; writing—review and editing, K.Y., M.T., A.N., H.A., Y.M. and J.-i.S.; visualization, T.M.; supervision, A.N., H.A., Y.M. and J.-i.S.; project administration, T.M., Y.M. and J.-i.S. All authors have read and agreed to the published version of the manuscript.

**Funding:** This research received no external funding.

**Institutional Review Board Statement:** The study was conducted in accordance with the Declaration of Helsinki and approved by the Institutional Review Board of Toyama University Hospital (protocol code R2019173; the approval date was 21 February 2020).

**Informed Consent Statement:** Patient consent was waived by the Toyama University Hospital Clinical Research Review Board because of the retrospective, observational nature of this study, but all the patients in the study had the opportunity to opt out.

**Data Availability Statement:** All the datasets are included within the article.

**Conflicts of Interest:** The authors declare no conflicts of interest.

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
