# Peer review of "Accelerated Fractionated Radiation Therapy for Localized Glottic Carcinoma"

_curroncol, doi:10.3390/curroncol31050198_

Round 1
Reviewer 1 Report
Comments and Suggestions for Authors
Dear authors,
The authors retrospectively analysed the results of accelerated irradiation for glottic carcinoma without lymph node metastasis. Twenty-nine patients were treated with dose fractions of 2.1-2.25 Gy with or without concurrent TS-1. The 3-year local control rate was 95.6%, comparable to previous reports. In terms of toxicity, serious adverse events were not common except for one patient who underwent tracheostomy for grade 4 laryngeal edema.
Accelerated radiotherapy for early glottic carcinoma has been widely studied, however reports of concurrent administration of TS-1 in this setting are scarce. In this regard, this article may be of interest to Current Oncology readers.
However, there are several critical issues that need to be addressed before this article can be accepted for publication in the journal.
1. Abstract: There are many grammatical errors. For example, the sentence "29 patients with N0 glottic carcinoma were treated..." makes no sense unless the passive voice is used. In addition, the dose fractionation and total dose should also be described in detail in the abstract.
2. Results: Although it is stated that a local recurrence was observed in patients with T2 disease, Table 2 shows that a recurrence was observed in a patient with T1 disease.
3. Results: The first three lines appear to have been inserted from the journal's instructions to authors. They should be deleted.
4. Methods: The description of the field size needs to be changed, e.g. 5 cm "x" 5 cm. The description of the dose and courses of TS-1 and vitamin A performed is inadequate. Despite the repeated mention of the benefit of shortening the total treatment time in the article, there is no description of the total treatment time for the patients analyzed in this study.
5. Results: Mixed use of mucositis and pharyngitis is confusing.
6. Results: The censor lines in Figure 1 should be more clearly delineated.
7. Discussion: Although several papers have been published on the treatment outcomes of radiotherapy with TS-1 for early glottic carcinoma, none of them are cited as references in this article. Comparison of the current results with these reports should be of paramount importance.
Regards,
Comments on the Quality of English Language
There are many grammatical errors in this paper.
The authors should correct their manuscript using proper services.
Reviewer 2 Report
Comments and Suggestions for Authors
The authors conducted a retrospective analysis to evaluate the effectiveness of accelerated fractionated radiation therapy in managing glottic carcinoma.
The topic is clinically interesting, and the study has merits for publication. The following comments are to improve the clarity of the manuscript.
1- The introduction is very concise, and the authors are advised to expand it to provide a summary of the current literature on accelerated fractionated radiation therapy. It is also recommended that details be provided on the global burden of glottic carcinoma.
2- The sample size is relatively small, but that is specifically related to the fact that the current manuscript describes the results from a single centre. This limitation needs to be carefully considered in the discussion.
3- The discussion can be improved by providing deeper insight by comparing AFRT and other radiation modalities.
4- The list of references doesn't cover all relevant studies. Please discuss Lacas, B., Bourhis, J., Overgaard, J., Zhang, Q., Grégoire, V., Nankivell, M., Zackrisson, B., Szutkowski, Z., Suwiński, R., Poulsen, M., O'Sullivan, B., Corvò, R., Laskar, S. G., Fallai, C., Yamazaki, H., Dobrowsky, W., Cho, K. H., Beadle, B., Langendijk, J. A., Viegas, C. M. P., … MARCH Collaborative Group (2017). Role of radiotherapy fractionation in head and neck cancers (MARCH): an updated meta-analysis. The Lancet. Oncology, 18(9), 1221–1237. https://doi.org/10.1016/S1470-2045(17)30458-8
Round 2
Reviewer 1 Report
Comments and Suggestions for Authors
The authors have revised the manuscript, but there is still one point that needs to be corrected.
Discussion: The authors do not currently use accelerated radiotherapy with concurrent TS-1 for patients with early glottic carcinoma, considering the risk of probably uncommon but severe late toxicity of laryngeal edema. Accelerated radiotherapy with concurrent chemotherapy is not considered to improve outcomes in patients with locally advanced head and neck cancer, as shown in the RTOG0129 trial. Yamazaki et al had shown in their phase III trial cited in the article that accelerated radiotherapy alone achieved favourable local control without increasing severe toxicity in the treatment of early glottic carcinoma. In view of the above, the author should emphasize the potential risk of combining TS-1 with accelerated radiotherapy. Also, the sentence in the discussion section "A comparative study with a larger number of cases of AFRT, normal fraction irradiation with TS-1, and AFRT with TS-1 should be considered" should be removed, as this type of study is unlikely to be performed in a randomised fashion in the future.
Round 3
Reviewer 1 Report
Comments and Suggestions for Authors
The authors corrected the manuscript according to my comments properly.